# Tumor-Infiltrating CD8-Positive T-Cells Associated with MMR and p53 Protein Expression Can Stratify Endometrial Carcinoma for Prognosis

**DOI:** 10.3390/diagnostics13121985

**Published:** 2023-06-06

**Authors:** Satoru Munakata, Takahiro Ito, Takuya Asano, Tsuyoshi Yamashita

**Affiliations:** 1Department of Pathology, Hakodate Municipal Hospital, 1-10-1 Minato-Cho, Hakodate 041-8680, Hokkaido, Japan; 2Department of Obstetrics and Gynecology, Hakodate Municipal Hospital, 1-10-1 Minato-Cho, Hakodate 041-8680, Hokkaido, Japan; takahiro-ito@hospital.hakodate.hokkaido.jp (T.I.); takuya-asano@hospital.hakodate.hokkaido.jp (T.A.); tyamashi@hospital.hakodate.hokkaido.jp (T.Y.)

**Keywords:** endometrial carcinoma, immunohistochemistry, classification, tumor-infiltrating lymphocytes, prognosis

## Abstract

Background: Inspired by the molecular classification of endometrial carcinoma (EC) proposed by The Cancer Genome Atlas Research Network (TCGA), we investigated tumor-infiltrating CD8-positive T-cell as well as DNA mismatch repair (MMR) protein and p53 protein expression, and we developed a new classification system for ECs to predict patients’ prognosis using immunohistochemical methods. Methods: The study included 128 patients with ECs who underwent surgery. Paraffin-embedded tissue sections of the tumor were stained using antibodies against MMR protein, p53, and CD8. Cases were stratified into four classes by a sequential algorithm. An immunohistochemical classification system for ECs (ICEC) was created, including HCD8, MMR-D, LCD8, and p53 LCD8. Results: In ICEC, 16 cases (12.5%), 27 cases (21.09%), 67 cases (52.34%), and 18 cases (14.06%) belonged to HCD8, MMR-D, LCD8, and p53 LCD8, respectively. ICEC did not show any correlation with clinical stage, lymphovascular space invasion, or lymph node metastasis. However, the p53 LCD8 class contained a significantly higher proportion of G3 ECs and serous carcinoma (*p* < 0.0001). ICEC showed prognostic significance in overall survival (OS) (*p* < 0.0001) and disease-free survival (DFS) (*p* < 0.0001). The class of p53 LCD8 showed the worst prognosis among the classes. Conclusions: ICEC classification is useful in predicting the prognosis of ECs.

## 1. Introduction

Endometrial carcinoma (EC) is the second most common gynecological carcinoma worldwide after cervical cancer, and it is the most common gynecological carcinoma in Japan [1]. Traditionally, endometrial carcinoma is divided into two types, type I and type II, as introduced by Bokhman [2] based on epidemiological and clinicopathological data. Type I tumors are low-grade endometrioid carcinomas with endometrial hyperplasia in the background and have a good prognosis. They are associated with excess estrogen, obesity, hypertension, hypercholesterolemia, glucose intolerance, and diabetes mellitus [2,3]. Type II tumors are high-grade tumors, including serous carcinoma and carcinosarcoma with an atrophic endometrium in the background, with poor prognosis, and they are not associated with excess estrogen and metabolic disturbances [2,3]. This dualistic model of Bokhman is useful in understanding endometrial carcinoma and managing patients with endometrial carcinoma [4]. However, this model is imperfect because there are carcinomas with ambiguous features that are difficult to classify. These include carcinomas with solid endometrioid architecture, glandular endometrioid architecture with a high nuclear grade, clear cells, and mixed epithelial components [5,6].

In 2013, The Cancer Genome Atlas Research Network (TCGA) proposed a comprehensive genomic and transcriptomic classification system for endometrial carcinoma [7]. The TCGA classification system is composed of four classes: (1) POLE (ultramutated) (POLEmut); (2) microsatellite instability (MSI) (hypermutated); (3) copy number low (endometrioid) (CN-low); and (4) copy number high (serous-like) (CN-high). This classification system is also well correlated with prognosis [7,8]. Although the TCGA classification system is useful for clinical purposes, it requires frozen material and molecular analysis. Based on the TCGA classification system, a more practical classification system for endometrial carcinoma using immunohistochemistry has been proposed by two groups [9,10,11,12]. Although these classification systems appear to be useful, the molecular analysis of *POLE* mutations, which remains a challenge in community hospital laboratories in Japan, is necessary in their models.

Tumor-infiltrating lymphocytes (TILs) are known to be associated with better prognosis in many tumors, including ECs [13,14,15]. Interestingly, ECs in the POLEmut and MSI classes are reported to have a significantly higher number of TILs in the TCGA classification [16,17,18,19,20].

Inspired by the TCGA classification and information about the clinical significance of TILs in EC, we aimed to develop a new prognostic classification system for EC using immunohistochemistry (ICEC), which can be easily utilized in the laboratories of community hospitals in Japan, where genomic mutation analysis is still uncommon.

## 2. Materials and Methods

### 2.1. Cases

Cases with a pathological diagnosis of EC were sought in the pathology database of Hakodate Municipal Hospital between 2009 and 2018. Formalin-fixed paraffin-embedded tissue (FFPE) blocks from patients who underwent hysterectomy were used. The pathological diagnosis of the specimens was reevaluated, and the eligibility of the specimens was determined by one of the authors (SM). In total, 136 cases were selected. As we aimed to determine the prognostic relevance of ECs, we selected the cases according to the TCGA study criteria [7]. In this regard, those with histology including clear cell carcinoma (*n* = 3), neuroendocrine carcinoma (*n* = 3), and carcinosarcoma (*n* = 2) were excluded from the study. One hundred and twenty-eight cases with endometrioid histology and serous histology were eligible for the study. The clinical records and follow-up data of the patients were obtained from the clinical database of Hakodate Municipal Hospital. All the patients had follow-up data. Clinicopathological characteristics are shown in Table 1. All cases were treated according to the standard clinical guidelines. The study was conducted according to the guidelines of the Declaration of Helsinki and approved by the institutional review board of Hakodate Municipal Hospital (IRB admission No.: 2021-12), and permission was obtained from all patients.

### 2.2. Immunohistochemistry

In each case, a representative FFPE block was selected for study. Three-μm FFPE specimens were cut and stained using Bond Max (Leica Biosystems K.K. Tokyo, Japan) or Bond-III (Leica Biosystems). Six monoclonal antibodies were used for the study, including p53 (DO-7, mouse monoclonal, ready to use, heat, Nichirei, Tokyo, Japan), antibodies for DNA mismatch repair (MMR) protein including MLH1 (ES05, mouse monoclonal, ×50, heat, Leica Biosystems), MSH2 (79H11, mouse monoclonal, ×150, heat, Leica Biosystems), MSH6 (PU29, mouse monoclonal, ×100, heat, Leica Biosystems), and PMS2 (M0R4G, mouse monoclonal, ×100, heat, Leica Biosystems), along with CD8 (C8/144B, mouse monoclonal, ready to use, heat, Nichirei). Immunohistochemical positivity for p53 was determined when 70% or more tumor cells were strongly stained in the nuclei or completely absent in staining for p53. Cases with a subclonal pattern of p53 expression were also determined to be positive according to our criteria, as stated by Singh et al. in their report [21]. MMR protein expression was considered aberrant if staining was entirely lost. If a subclonal pattern was observed, the patient was also determined to be MMR-deficient according to the results reported by Stelloo et al. [22]. For CD8-positive T-cell (TC) counting, 4 randomly selected areas were used for analysis. Digital images were obtained using a NikonDS-Fi3 digital camera and the NIS-Elements Lite software Ver. 1.00 with a 20× objective lens. Four of the 0.33 mm^2^ areas, amounting to 1.35 mm^2^ in total, were used to count CD8-positive TCs. The intra-tumoral and peri-tumoral infiltration of CD8-positive TCs was manually counted using counting software ver. 2.71 (katikati2: GTSOFT) by one of the authors (SM). Intra-tumoral CD8-positive TC infiltration with up to 400 cells was designated as CD8 TIL-low. Intra-tumoral CD8-positive TC infiltration with 400 cells or more in total was designated as CD8 TIL-high. Accordingly, peri-tumoral CD8 infiltration up to 400 cells was designated as peri CD8-low, and peri-tumoral CD8 infiltration with 400 cells or more was designated as peri CD8-high.

### 2.3. Immunohistochemical Classification

The algorithm for the immunohistochemical classification of endometrial carcinoma (ICEC) is shown in Figure 1. First, the MMR protein deficiency status was checked. Cases with MMR protein deficiency were classified as MMR-D. Next, the intra-tumoral CD8-positive TC count was evaluated. Cases with CD8 TIL-high were classified into the HCD8 class, regardless of the p53 staining results. After the selection of the CD8 TIL status, cases considered p53-positive and CD8 TIL-low were classified into the p53 LCD8 class. The remaining cases that were MMR-proficient, p53-negative, and CD8 TIL-low were classified as into the LCD8 class.

### 2.4. Statistics

Statistical analysis was performed using the Statcel Ver. 3 software (OMS, Japan). The association between the ICEC class and age was tested using the Kruskal–Wallis test. Associations between the ICEC class and clinical stage, histological grade, histological subtype, lymphovascular space invasion (LVSI), and lymph node metastasis were calculated using the chi-square for independence test with an m × n contingency table. The correlation between intra-tumoral CD8-positive TC and peri-tumoral CD8-positive TC infiltration was calculated using the Pearson correlation coefficient. The difference in the number of infiltrated CD8-positive TCs between intra-tumoral and peri-tumoral tissue was tested by Welch’s *t* test. Survival curves were calculated using the Kaplan–Meier method with the log-rank test.

## 3. Results

### 3.1. Immunohistochemical Results

Positive staining for the antibodies were observed in 23 cases (18.0%), 112 cases (87.5%), 126 cases (98.4%), 123 cases (96.1%), and 106 cases (82.8%) out of 128 cases for p53, MLH1, MSH2, MSH6, and PMS2, respectively. CD8 TIL-high was found in 23 cases (18.0%). Peri CD8-high was found in 15 cases (11.7%). There was a weak positive correlation between intra-tumoral CD8-positive TCs and peri-tumoral CD8-positive TCs (r = 0.25, *p* = 0.004). Representative images of the ICEC classes are shown in Figure 2, Figure 3, Figure 4, Figure 5 and Figure 6.

### 3.2. Clinicopathological Characteristics of ICEC

Associations between the classes determined by ICEC and clinicopathological characteristics are shown in Table 2. The mean age of the patients with p53 LCD3 was significantly higher among the classes (*p* = 0.001). The classes of ICEC were not significantly associated with the clinical stages. Histological analysis showed that a significantly higher number of serous carcinomas were observed in the p53 LCD8 class (*p* < 0.0001). No significant association was observed between the ICEC classes and LVSI or lymph node metastasis.

### 3.3. Intra-Tumoral and Peri-Tumoral CD8-Positive TCs

The numbers of intra-tumoral and peri-tumoral CD8-positive TCs are shown in Table 3. These were compared between ICEC classes. The number of CD8-positive TCs in the HCD8 class was the highest among the classes of ICEC. The number of CD8-positive TCs in the HCD8 class was also significantly higher than that in the MMR-D class (*p* = 0.025).

### 3.4. Clinical Outcomes and ICEC

Clinical outcomes were investigated using multiple parameters (Table 4). In the association of the classes defined by ICEC, the class of p53 LCD8 showed the worst prognosis in terms of overall survival (OS) (*p* < 0.0001) and disease-free survival (DFS) (*p* < 0.0001) (Figure 7). In contrast, the HCD8 class showed excellent prognosis in OS and DFS.

The International Federation of Gynecology and Obstetrics (FIGO) grade showed significant prognostic significance in OS (*p* < 0.0001) and DFS (*p* < 0.0001). The FIGO stage also showed prognostic significance in OS (*p* < 0.0001) and DFS (*p* = 0.0002).

p53-positive tumors showed worse prognosis in OS (*p* < 0.0001) and DFS (*p* < 0.0001). Among the cases with p53-positive tumors (*n* = 23), cases with CD8 TIL-high showed a tendency toward better OS (*p* = 0.07), while these cases showed significantly better DFS (*p* = 0.03) (Figure 8).

Tumors with intra-tumoral CD8 TIL-high showed a better prognosis in OS (*p* = 0.04) and DFS (*p* = 0.03). However, peri-tumoral CD8 TIL did not show a significant prognostic difference.

LVSI status showed a poorer prognosis in OS (*p* = 0.01) and DFS (*p* = 0.001). In the cases with an available lymph node metastatic status (*n* = 112), those with lymph node metastasis exhibited worse prognosis in OS (*p* = 0.0004) and DFS (*p* = 0.008).

## 4. Discussion

Since the introduction of TCGA’s comprehensive genomic classification system for endometrial carcinoma, many studies have confirmed the usefulness of this molecular classification system in clinical practice [7,8,19,23,24]. This molecular classification system differs from Bohkman’s dualistic classification system. In contrast to the incomplete clinicopathological classification of Bohkman, the molecular analysis of TCGA shows the possibility to classify endometrial carcinoma in terms of clinical usefulness, because some of the tumors are difficult to classify histologically when predicting prognosis [5,25]. While histological cell types are important prognostic parameters of endometrial carcinoma, the reproducibility of histological cell types is relatively low and consensus histological diagnoses require many immunohistochemical markers [6]. McConechy et al. proposed to refine the classification of endometrial carcinomas using the mutation profiles of nine genes, namely *ARID1A*, *PPP2R1A*, *PTEN*, *PIK3CA*, *KRAS*, *CTNNB1*, *TP53*, *BRAF*, and *PPP2R5C* [25]. They stated that the molecular profile of the tumor was useful as an adjunct to morphological classification and could serve as an aid in the classification of ECs. However, they still intended to classify ECs via two-tiered classification [25]. The TCGA classification system proposed a new means of classification to determine the biological characteristics and behavior of the tumor as indicated by the patient’s prognosis, to provide a more appropriate choice of treatment modalities for each patient. The four distinct classes of the TCGA classification system have particular molecular profiles and prognostic value. Traditional histological classification cannot exactly predict each class of the TCGA classification system. In recent studies of the TCGA classification system, four molecular and immunohistochemical classes were found to contain a variety of histological types [26,27]. In our study, each ICEC class also had a variety of histological grades, with a significant difference between the classes (Table 2). In particular, the p53 LCD8 class, which showed the worst prognosis, had a substantial number of cases with G3 endometrioid carcinoma and serous carcinoma, although even this class had a substantial number of cases with G1 and G2 endometrioid carcinoma. In contrast, one quarter of the HCD8 class, which exhibited excellent prognosis, showed G3 endometrioid carcinoma. Therefore, the histological classification and grade alone cannot predict ICEC classification, as observed in the TCGA classification system.

POLEmut and MSI tumors of TCGA classification are reported to be highly associated with an increased number of tumor-infiltrating TCs [16,17,18,19,20,28]. A mutation of *POLE*, DNA polymerase ε, which has polymerase activity and 3′-5′ exonuclease activity, along with MMR deficiency, will cause a high tumor mutation burden, resulting in the accumulation of mutated genes in cells and the production of tumor neoantigens. It has been reported that an increased number of tumor-infiltrating TCs is associated with increased neoantigen production and the expression of programmed death receptor-1 (PD-1) and its ligand, PD-L1, a target of immune checkpoint blockade, in POLEmut and MSI tumors [16,18,20,28]. In our study, the number of intra-tumoral CD8-positive TCs in the HCD8 and MMR-D classes was significantly higher than that in the LCD8 and p53 LCD8 classes. Based on these observations, the HCD8 class has potential to indicate the number of POLEmut tumors in the TCGA classification system, although POLE mutation analysis was not conducted in this study.

In the TCGA classification system, the study materials were endometrioid, serous, and mixed carcinomas [7]. This study did not include clear cell carcinoma. In our study, only endometrioid carcinoma and serous carcinoma were included, while other histologies, including clear cell carcinoma, were excluded according to the TCGA classification.

A higher number of TILs, especially tumor-infiltrating CD8-positive TCs, is known to be related to better prognosis in endometrial carcinoma [13,14]. In our study, tumors with CD8 TIL-high status showed significantly better prognosis than those with CD8 TIL-low status in OS (*p* = 0.04) and DFS (*p* = 0.03), as shown in previous studies (Table 4) [13,14].

MSI was evaluated by immunohistochemistry as a surrogate marker for molecular analysis. It was reported that MMR deficiency was concordant with MSI in 94% of cases [22]. Although the assessment of MMR protein expression by immunohistochemistry is difficult, one study showed interobserver agreement of 92% [29]. Stelloo et al. reported that they observed the subclonal expression of MMR proteins in <3% of cases. They stated that the subclonal expression of MMR proteins should be classified as MMR deficiency because most cases with subclonal expression showed MSI-H [22]. In our study, the subclonal expression of MLH1 was observed in 2 out of 27 cases of MMR-D tumors (7%). Because these tumors were also deficient in PMS2 expression, they were classified as MMR-D tumors.

Immunohistochemistry for p53 is well known to represent *TP53* mutation. Singh et al. reported that the immunohistochemical evaluation of p53 was concordant with *TP53* mutation in 92.3% of cases [21]. They observed that the subclonal expression of p53 belonging to the POLEmut and MMR-deficient classes did not have *TP53* mutation, while those belonging to the *POLE* wild type and MMR-proficient classes showed *TP53* mutation in four of five cases (80%) [21]. In our study, subclonal p53 expression was observed in four cases (two MMR-D and two LCD8). Because mutation analysis for *TP53* was not available in our study, the expression of p53 in these cases was classified according to our study criteria, i.e., strongly positive when ≥70% or complete absence. Tumors with aberrant p53 expression were also found in the POLEmut class of the TCGA classification system [7,30]. Considering that the POLEmut class contains tumors with a high number of intra-tumor lymphocytes and aberrant p53 expression, the tumors with intra-tumoral TIL-high with aberrant p53 expression observed in this study might reflect tumors of the POLEmut class in the TCGA classification system; however, this association could not be proven due to the lack of mutation analysis. Instead, we developed the new classification system employing the immunohistochemical analysis of intra-tumoral CD8-positive TC infiltration, MMR status, and p53 status. In our study, CD8-positive TIL-high tumors showed a better prognosis than CD8-positive TIL-low tumors, even in tumors with aberrant p53 expression with OS (*p* = 0.07) and DFS (*p* = 0.03) (Figure 8). Our observations showed the prognostic importance of CD8-positive TCs, playing an antitumor immunological protective role irrespective of the aberrant p53 expression of tumors. Notably, this was proven by the fact that the HCD8 class showed excellent prognosis.

The practical application of TCGA classification using paraffin-embedded tissue was proposed by two groups [9,10,11,12]. Talhouk et al. proposed the ProMisE system using immunohistochemistry for MMR and p53 protein expression, along with molecular analysis of *POLE* mutations [9,10]. They determined four groups of POLE EDM, MMR IHC abn, p53 wt, and p53 abn for POLEmut, MSI hypermutated, CN-low, and CN-high in the TCGA classification system, respectively. In their study, 9–9.4%, 20–29%, 43.6–45%, and 18–27% of cases were allocated to POLE EDM, MMR IHC abn, p53 wt, and p53 abn, respectively [9,10]. In our study, 12.5%, 21.09%, 52.34%, and 14.06% of cases were allocated to HCD8, MMR-D, LCD8, and p53 LCD8 (Table 2). This is very similar to their study. In this regard, our classification system might be useful to determine the biological behavior of tumors.

Both the ProMisE system and the PORTEC trial showed the practical usefulness of molecular and immunohistochemical classifiers in managing patients with endometrial cancer. de Biase et al. compared the European Society of Gynecological Oncology/European Society for Radiotherapy and Oncology/European Society of Pathology endometrial cancer risk classification system (ESGO/ESTRO/ESP2016) with immuno-molecular analysis incorporating ESGO/ESTRO/ESP2020, to evaluate the prognostic impact in endometrial cancer. They found that the new classification system, including the analysis of *POLE* mutation, MMR, and p53 status, was more suitable to stratify ECs for prognosis [24]. However, mutation analysis of *POLE* is the only obstacle preventing these classifiers from becoming standard surrogates for TCGA classification worldwide, because there are many countries where molecular analysis is still challenging to apply. Our ICEC solely uses immunohistochemical analysis, which can be applied even in the laboratories of community hospitals in Japan.

Our study showed the possibility that immunohistochemical analysis can stratify ECs for prognosis without molecular analysis. In particular, we could separate the population with good prognosis even in p53-positive ECs. Interestingly, the study by Meng et al. reported that *POLE* mutant tumors showed significantly better prognosis in grade 3 endometrioid carcinoma [31]. Their study showed the importance of separating tumors with relatively benign behavior from tumors that had previously been classified as a high-risk group according to histological and immunohistochemical parameters. Our data are in line with the study of Meng et al. with respect to finding relatively benign ECs in tumors that were previously classified as high-risk ECs. Our developed ICEC can provide useful information for clinicians who treat patients with ECs. In our ICEC classification, ECs of HCD8 are considered as a benign group and ECs with p53 LCD8 as an aggressive group. ECs with MMR-D and LCD8 are considered as an intermediate behavior group.

However, it would be premature to draw conclusions because of the relatively small number of samples and shorter follow-up periods in the study. Therefore, further investigation is necessary to confirm our data. We hope that our immunohistochemical classifier will aid in the clinical management of EC in the future.

In conclusion, we showed the clinically relevant classification of endometrial carcinoma when the analysis of MMR and p53 protein expression was combined with the assessment of tumor-infiltrating CD8-positive TCs, inspired by the concepts of TCGA. We created a prognosis-associated new classification system only using immunohistochemistry and showed distinct class stratification with prognostic significance. Practically, the ICEC must be applied to biopsy specimens to provide clinicians with useful information before they choose therapeutic modalities. Further investigation is necessary to prove the usefulness of our new classifier in treating patients with endometrial carcinoma.

## Figures and Tables

**Figure 1 diagnostics-13-01985-f001:**
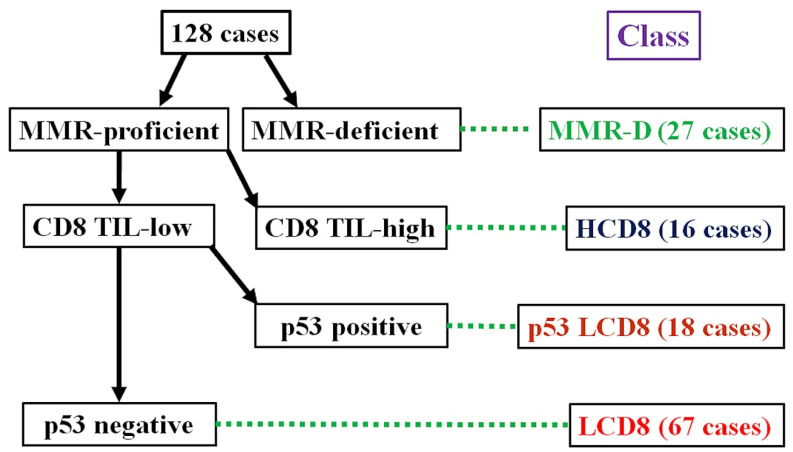
Algorithm for immunohistochemical classifier of endometrial carcinoma (ICEC).

**Figure 2 diagnostics-13-01985-f002:**
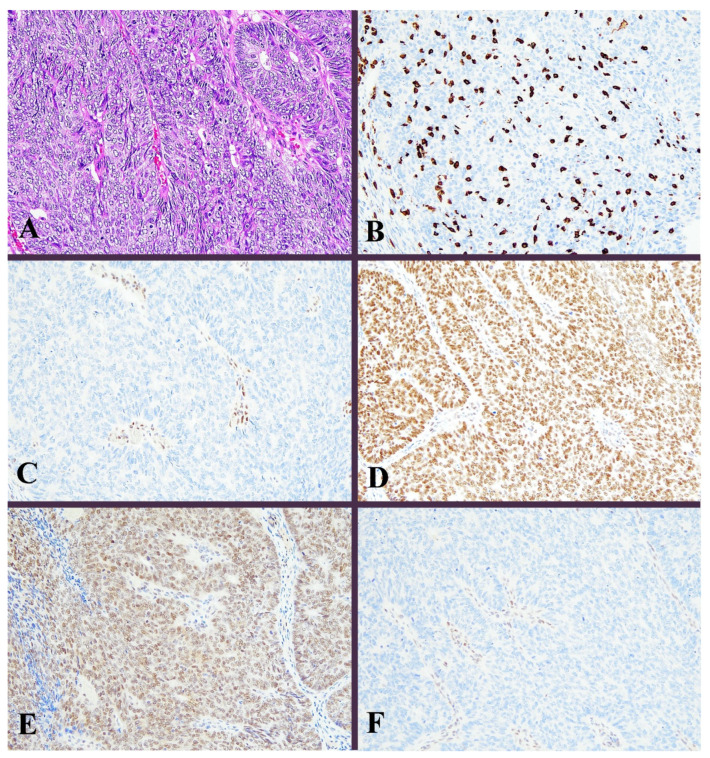
Histological image for a case classified as MMR-D. (**A**) This case was diagnosed as endometrioid carcinoma (G3). Solid growth of tumor cells contained many tumor-infiltrating lymphocytes. (**B**) CD8-stained T-lymphocytes massively infiltrated into the tumor nests. (**C**) MLH1, (**D**) MSH2, (**E**) MSH6, and (**F**) PMS2. Staining of MLH1 and PMS2 was not observed, in contrast to positive staining in lymphocytes. ((**A**): H&E staining, ×20, (**B**–**F**): Immunohistochemistry, ×20).

**Figure 3 diagnostics-13-01985-f003:**
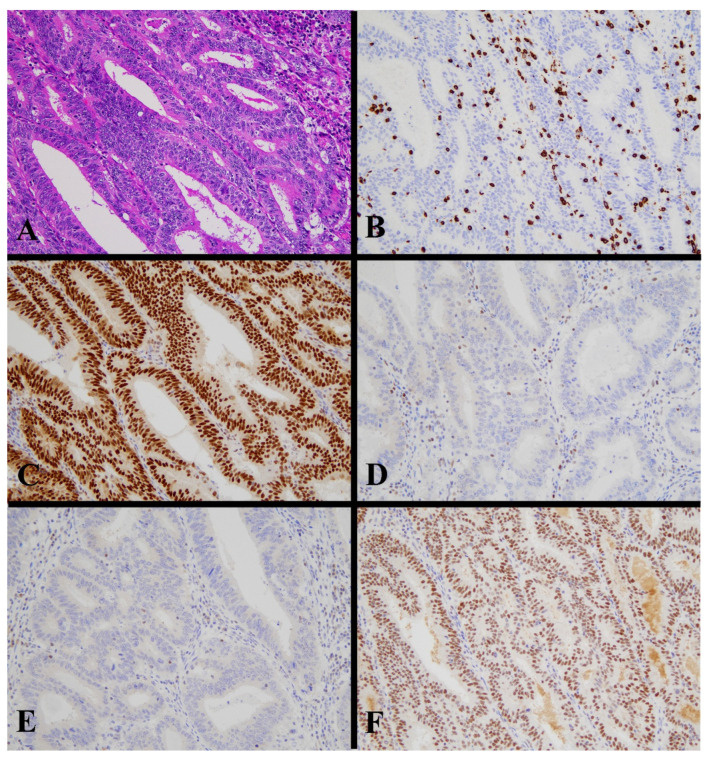
Histological image for a case classified as MMR-D. (**A**) This case was diagnosed as endometrioid carcinoma (G2). Tumor cells contained many tumor-infiltrating lymphocytes. (**B**) CD8-stained T-lymphocytes massively infiltrated into the tumor nests. (**C**) MLH1, (**D**) MSH2, (**E**) MSH6, and (**F**) PMS2. Staining of MLH2 and MSH6 was not observed, in contrast to positive staining in lymphocytes. ((**A**): H&E staining, ×20, (**B**–**F**): Immunohistochemistry, ×20).

**Figure 4 diagnostics-13-01985-f004:**
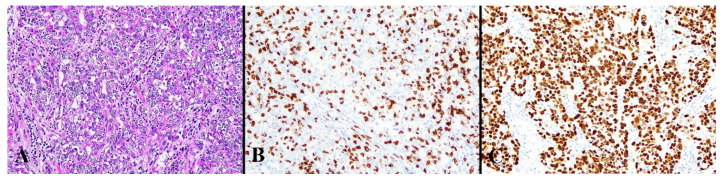
Histological image for a case classified as HCD8. (**A**) This case was diagnosed as endometrioid carcinoma (G3). (**B**) CD8-stained T-lymphocytes massively infiltrated into the tumor nests. (**C**) This case was positive for p53. ((**A**): H&E staining, ×20, (**B**,**C**): Immunohistochemistry, ×20).

**Figure 5 diagnostics-13-01985-f005:**
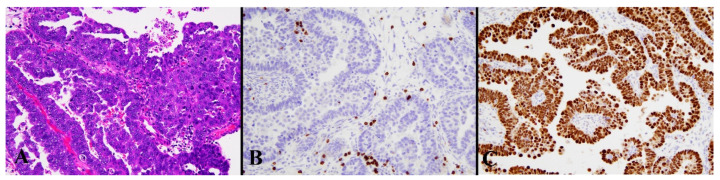
Histological image for a case classified as p53 LCD8. (**A**) This case was diagnosed as serous carcinoma. (**B**) CD8-positive T-lymphocyte infiltration is sparse. (**C**) p53 is strongly positive in 70% or more of tumor cells. ((**A**): H&E staining, ×20, (**B**,**C**): Immunohistochemistry, ×20).

**Figure 6 diagnostics-13-01985-f006:**
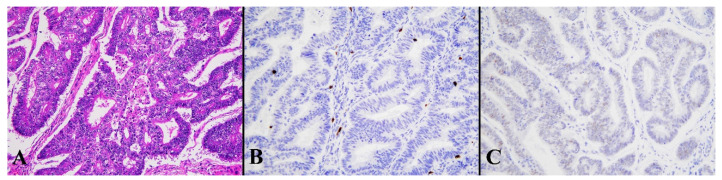
Histological image for a case classified as LCD8. (**A**) This case was diagnosed as endometrioid carcinoma (G1). (**B**) CD8-positive T-lymphocyte infiltration is sparse. (**C**) This case was p53-negative. ((**A**): H&E staining, ×20: (**B**,**C**): Immunohistochemistry, ×20).

**Figure 7 diagnostics-13-01985-f007:**
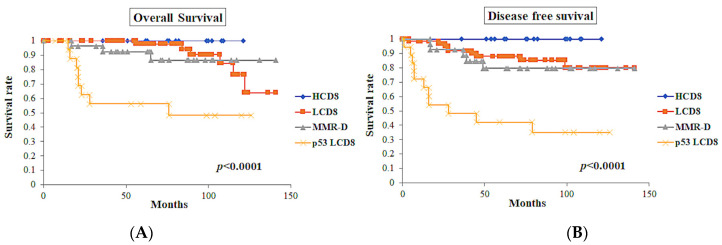
Survival curves of the classes defined by ICEC (*n* = 128). (**A**) Overall survival curves of the classes. (**B**) Disease-free survival curves of the classes. The class of p53 LCD8 showed the worst overall survival (*p* < 0.0001) and disease-free survival (*p* < 0.0001).

**Figure 8 diagnostics-13-01985-f008:**
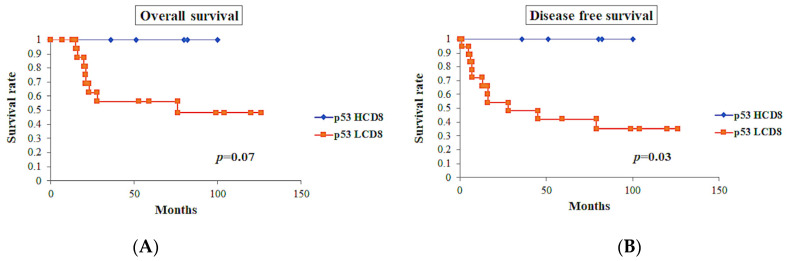
Survival curves in p53-positive tumors (*n* = 23). (**A**) Overall survival of the cases with CD8 TIL-high showed tendency toward good prognosis, but did not show significant differences (*p* = 0.07). (**B**) Disease-free survival of the cases with CD8 TIL-high showed significantly better prognosis (*p* = 0.03).

**Table 1 diagnostics-13-01985-t001:** Clinicopathological characteristics of the cases.

**Mean Age ± SD**	59.27 ± 12.0 years (range: 31–85 years)
**Histology (*n* = 128)**	
Endometrioid carcinoma (G1)	73 (57%)
Endometrioid carcinoma (G2)	32 (25%)
Endometrioid carcinoma (G3)	17 (13.3%)
Serous carcinoma	6 (4.7%)
**Clinical stage (FIGO *) (*n* = 128)**	
IA	74 (57.8%)
IB	22 (17.2%)
II	5 (3.9%)
III	23 (18.0%)
IV	4 (3.1%)
**LVSI ****	15 (11.7%)
**Lymph node metastasis (*n* = 112)**	22 (19.6%)
**Follow up**	72.0 ± 34.5 months (range: 3–141 months)

* FIGO: The International Federation of Gynaecology and Obstetrics, ** LVSI: lymphovascular space invasion.

**Table 2 diagnostics-13-01985-t002:** Clinicopathologic characteristics in ICEC *.

Characteristics	Class				
	HCD8	MMR-D	LCD8	p53 LCD8	*p*
Age (mean ± SD)	59.7 ± 13.2 (range: 38–83)	60.5 ± 7.8 (range: 43–77)	56.5 ± 12.5 (range: 31–82)	69.2 ± 9.3 (range: 53–85)	0.001
Clinical stage (FIGO)					
IA	9 (56.25%)	16 (59.26%)	41 (61.19%)	8 (59.1%)	NS **
IB	3 (18.75%)	3 (11.11%)	13 (19.4%)	3 (16.67%)	
II	1 (6.25%)	1 (3.7%)	3 (4.48%)	0 (0%)	
III	3 (18.75%)	7 (25.93%)	7 (10.45%)	6 (33.33%)	
IV	0 (0%)	0 (0%)	3 (4.48%)	1 (5.56%)	
Histology (grade)					
Endometrioid (G1)	6 (37.5%)	12 (44.44%)	52 (77.61%)	3 (16.67%)	<0.0001
Endometrioid (G2)	6 (37.5%)	10 (37.04%)	12 (17.91%)	4 (22.22%)	
Endometrioid (G3)	4 (25.0%)	5 (18.52%)	3 (4.48%)	5 (27.78%)	
Serous	0 (0%)	0 (0%)	0 (0%)	6 (33.33%)	
LVSI §	1 (6.25%)	6 (22.22%)	5 (7.46%)	3 (16.67%)	NS
Lymph node metastasis (*n* = 112)	2/16 (12.5%)	6/23 (26.09%)	9/58 (15.52%)	5/15 (33.33%)	NS
Total	16 (12.5%)	27 (21.09%)	67 (52.34%)	18 (14.06%)	

* ICEC: immunohistochemical classifier for endometrial carcinoma, ** NS: not significant, § LVSI: lymphovascular space invasion.

**Table 3 diagnostics-13-01985-t003:** The numbers of intra-tumoral and peri-tumoral CD8 TCs * among ICEC **.

Characteristics	Class				
	HCD8	MMR-D	LCD8	p53 LCD8	*p*
Intra-tumoral CD8 TCs ‡	833.1 (±544)	413.7 (±609)	112.1 (±85)	90.4 (±78)	*p* < 0.0001
Peri-tumoral CD8 TCs	270.3 (±145)	229.4 (±233)	192 (±162)	164.2 (±147)	NS §

* TCs: T-cells, ** ICEC: immunohistochemical classifier for endometrial carcinoma, ‡: mean, with SD in parentheses, § NS: not significant.

**Table 4 diagnostics-13-01985-t004:** Prognostic significance according to the tumor characteristics *.

	OS **	DFS §
All cases (*n* = 128)		
ICEC	*p* < 0.0001	*p* < 0.0001
FIGO grade	*p* < 0.0001	*p* < 0.0001
FIGO stage	*p* < 0.0001	*p* = 0.0002
p53 status	*p* < 0.0001	*p* < 0.0001
Intra-tumoral CD8 TIL	*p* = 0.04	*p* = 0.03
Peri-tumoral CD8 TIL	*p* = 0.85	*p* = 0.55
LVSI ‡	*p* = 0.01	*p* = 0.001
Lymph node metastasis (*n* = 112)	*p* = 0.0004	*p* = 0.008

* *n* = 128 except cases having lymph node metastatic status (*n* = 112), ** OS: overall survival, § DFS: disease-free survival, ‡ LVSI: lymphovascular space invasion.

## Data Availability

All relevant data are provided within the paper.

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
