# Peer review of "Tumor-Infiltrating CD8-Positive T-Cells Associated with MMR and p53 Protein Expression Can Stratify Endometrial Carcinoma for Prognosis"

_diagnostics, 2023, doi:10.3390/diagnostics13121985_

Round 1
Reviewer 1 Report
The article is original research.
It is well written, the figures and diagrams are relevant to the topic.
The authors should increase the number of references.
Additional comments from reviewer:
The main idea addressed by the authors is a new molecular classification system on endometrial cancer with tumor-infiltrating CD-8 positive cells combined with DNA MMR protein and p53 protein expression.
I think that this specific manuscript is very interesting and innovative and does add new data in science.
I have observed in other publications that this specific topic has not been fully analyzed.
The topic is relevant to the field, and it does fill in gaps in the field.
I have already mentioned in my first evaluation that the number of the references are relevant but the number should be increased. In its current edition it is not enough.
The figures-tables and the pictures are quite relevant and detailed.
Regarding the conclusion it is precise and descriptive.
Minor revision would adequate.
Author Response
The article is original research.
It is well written, the figures and diagrams are relevant to the topic.
The authors should increase the number of references.
⇒ Thank you for your kind suggestion. We increased the number of references.
Additional comments from reviewer:
The main idea addressed by the authors is a new molecular classification system on endometrial cancer with tumor-infiltrating CD-8 positive cells combined with DNA MMR protein and p53 protein expression.
I think that this specific manuscript is very interesting and innovative and does add new data in science.
I have observed in other publications that this specific topic has not been fully analyzed.
The topic is relevant to the field, and it does fill in gaps in the field.
I have already mentioned in my first evaluation that the number of the references are relevant but the number should be increased. In its current edition it is not enough.
The figures-tables and the pictures are quite relevant and detailed.
Regarding the conclusion it is precise and descriptive.
Minor revision would adequate.
⇒ Thank you very much for your warmful comment. We revised the article with expanded volume and increased number of references.
Reviewer 2 Report
Munakata et al. in this work proposed a new way of classifying endometrial carcinoma by using immunohistochemical staining against MMR, CD8 and p53. Based on the result of the staining, they divided 128 cases of EC into four classes and showed that the group with positive p53 and low CD8 has the worst prognosis while CD8 positive TIL-high group showed favorable prognosis. This new classification of EC may be clinically relevant and should be of interest to the readers.
Author Response
Munakata et al. in this work proposed a new way of classifying endometrial carcinoma by using immunohistochemical staining against MMR, CD8 and p53. Based on the result of the staining, they divided 128 cases of EC into four classes and showed that the group with positive p53 and low CD8 has the worst prognosis while CD8 positive TIL-high group showed favorable prognosis. This new classification of EC may be clinically relevant and should be of interest to the readers.
⇒ Thank you very much for your kind comment. We also revised the article with expanded volume and increased number of references.
Reviewer 3 Report
The manuscript titled, “Tumor-infiltrating CD8 positive T-cells associated with MMR and p53 protein expression can stratify endometrial carcinoma for prognosis” by Munakata et al is indeed very interesting and novel. I recommend this manuscript for publication. I have one minor suggestion; the authors should include the translational impact and shortcomings in this study in a separate paragraph in discussion section as that can be useful to a potential reader.
Author Response
The manuscript titled, “Tumor-infiltrating CD8 positive T-cells associated with MMR and p53 protein expression can stratify endometrial carcinoma for prognosis” by Munakata et al is indeed very interesting and novel. I recommend this manuscript for publication. I have one minor suggestion; the authors should include the translational impact and shortcomings in this study in a separate paragraph in discussion section as that can be useful to a potential reader.
⇒ Thank you for your kind suggestion. We have revised the manuscript including translational impact and shortcomings in this study in the discussion section.